# Efficacy of Continuous Glucose Monitoring on Glycaemic Control in Pregnant Women with Gestational Diabetes Mellitus—A Systematic Review

**DOI:** 10.3390/jcm11102932

**Published:** 2022-05-23

**Authors:** Agata Majewska, Paweł Jan Stanirowski, Mirosław Wielgoś, Dorota Bomba-Opoń

**Affiliations:** 1st Department of Obstetrics and Gynaecology, Medical University of Warsaw, 02-015 Warsaw, Poland; stanirowski@gmail.com (P.J.S.); miroslaw.wielgos@wum.edu.pl (M.W.); dorota.bomba-opon@wum.edu.pl (D.B.-O.)

**Keywords:** gestational diabetes mellitus, continuous glucose monitoring, self-monitoring of blood glucose, hyperglycaemia, hypoglycaemia

## Abstract

Gestational diabetes mellitus (GDM) is one of the most common complications of pregnancy, affecting up to 14% of pregnant women. The population of patients with risk factors of GDM is increasing; thus, it is essential to improve management of this condition. One of the key factors affecting perinatal outcomes in GDM is glycaemic control. Until recently, glucose monitoring was only available with self-monitoring of blood glucose (SMBG). However, nowadays, there is a new method, continuous glucose monitoring (CGM), which has been shown to be safe in pregnancy. Since proper glycaemia assessment has been shown to affect perinatal outcomes, we decided to perform a systematic review to analyse the role of CGM in glycaemic control in GDM. We conducted a web search of the MEDLINE, EMBASE, Cochrane Library, Scopus, and Web of Science databases according to the PRISMA guidelines. The web search was performed by two independent researchers and resulted in 14 articles included in the systematic review. The study protocol was registered in the PROSPERO database with registration number CRD42021289883. The main outcome of the systematic review was determining that, when compared, CGM played an important role in better glycaemic control than SMBG. Furthermore, glycaemic control with CGM improved qualification for insulin therapy. However, most of the articles did not reveal CGM’s role in improving neonatal outcomes. Therefore, more studies are needed to analyse the role of CGM in affecting perinatal outcomes in GDM.

## 1. Introduction

Gestational diabetes mellitus (GDM) is the most common complication of pregnancy, with an incidence rate of up to 14% of all pregnant women [1]. Over a long period of time, it was defined as any degree of glucose intolerance with the onset or first recognition during pregnancy [2,3]. However, now it is debated whether this definition is appropriate due to its limitations, including imprecise information about diagnostic thresholds for GDM [4]. The population of patients with risk factors for GDM is continuously increasing, thus, it is essential to improve the management of GDM [5]. It is believed that glycaemic control plays a major role in the proper treatment of GDM [6,7]. Until recently glycaemic control in GDM was mainly based on the self-monitoring of blood glucose (SMBG) [8]. However, the main inconveniences of this method are multiple finger-pricking for a single glycaemia measurement and intermittent checking of glucose levels, which might lead to poor patient compliance [9]. Recently, a new method for glycaemic control was introduced, namely, continuous glucose monitoring (CGM) [10]. The method uses a subcutaneous sensor to collect the glycaemia results. The main benefit of CGM is that, after insertion, the system analyses the actual glycaemia constantly without any additional invasive procedure [11]. An important advantage of CGM is the evaluation of time the patient spends in normoglycaemia. It is called time-in-range, and it is defined as the percentage of time in which glycaemia is in reference range [12]. It is believed that time-in-range is a more accurate outcome to assess the patient’s compliance.

There are ongoing debates about what type of glycaemia measurement method is the most effective for pregnant women diagnosed with GDM. It is hypothesized that CGM is superior to SMBG, but due to the high price of the device and a lack of reimbursement for GDM in many countries, it is not used as the method of choice [7,8]. 

The aim of this systematic review is to assess the efficacy of continuous glucose monitoring on glycaemic control in pregnant women with GDM. In addition, this review will focus on the need for pharmacological treatment and perinatal outcomes in the population of patients using CGM.

## 2. Materials and Methods

### 2.1. Search Strategy and Selection Criteria

We conducted a systematic web search in the MEDLINE, EMBASE, Cochrane Central Register of Controlled Trials (CENTRAL), Scopus, and Web of Science databases according to the Preferred Reporting Items for Systematic Reviews and Meta-Analyses (PRISMA) guidelines. The systematic review has been registered in the International Prospective Register of Systematic Reviews (PROSPERO) registry (CRD42021289883). The keywords utilized for the research were: continuous glucose monitoring, flash glucose monitoring, and gestational diabetes mellitus. The time frame of the research was from database inception date to November 2021. The inclusion criteria were: randomized controlled trials and observational studies, and human studies in English. The exclusion criteria were types of studies other than the inclusion criteria, animal studies, and studies in different languages than English (Table 1). 

Following the initial screening, publications were analysed further by title and abstract to exclude studies that did not meet the inclusion criteria. After initial selection, the remaining full articles were screened to assess the final number of eligible publications included to the systematic review. Two of the authors independently evaluated all retrieved studies against the eligibility criteria and, in cases of differing opinion, the publication was discussed with the third author. 

Due to heterogeneity in terms of continuous glucose monitoring devices, study duration, and number of patients among the included articles, no meta-analysis was performed. 

### 2.2. Data Analysis

Data were extracted independently by two researchers. The following data were extracted: type of article, year of publication, type of continuous glucose monitoring, number of patients included in the study, fasting, postprandial and nocturnal glycaemia, time in range, qualification for insulin therapy, incidence of severe nocturnal hypoglycaemia, glycosylated haemoglobin concentration (HbA1c), gestational weight gain, newborn birth weight, and other neonatal outcomes. 

### 2.3. Outcomes

The main outcome was glycaemic control (fasting, postprandial and nocturnal glycaemia). Several secondary outcomes were also investigated, including: qualification to insulin therapy, incidence of severe nocturnal hypoglycaemia, HbA1c, gestational weight gain, newborn birth weight, and other neonatal outcomes.

## 3. Results

A total of 435 articles were identified through a systematic review of the literature (Figure 1). 

After initial screening, 172 duplicates were excluded and 263 titles and abstracts were screened further for eligibility criteria, leaving a total of 51 full-text publications. Review of the full-text articles resulted in 37 studies being excluded from further assessment. A total of 14 remaining publications were included in the final analysis of this systematic review (Table 2). 

### 3.1. Glycaemic Control 

#### 3.1.1. Hyperglycaemia

In five studies, it was found that CGM is better at detecting episodes of hyperglycaemia as compared to SMBG [7,9,12,15,16]. In two studies, it was found that CGM detected more hyperglycaemic events than SMBG [9,15]. However, Afandi et al. demonstrated that the incidence rate of hyperglycaemia in all patients included in the study reached 5.65% using CGM versus 14.2% using SMBG (*p* < 0.05) [7]. The incidence of hyperglycaemia above 180 mg/dL in the CGM and SMBG groups was estimated to be <1.0% and 2% of all readings, respectively (*p* < 0.05). In another prospective study, hyperglycaemic events were analysed further, and the result was that, in the CGM group, the duration of time spent in hyperglycaemia was shorter than in the SMBG group [12]. One study found that CGM is a better detector of nocturnal hyperglycaemia than SMBG [16]. On the other hand, three studies described no statistical difference between the SMBG and CGM groups in detecting glycaemia above the reference range [11,13,17].

#### 3.1.2. Hypoglycaemia

We found eight articles about incidences of hypoglycaemia [6,7,9,11,12,13,14,15]. In most studies, the outcome was that CGM detects a higher number of hypoglycaemia episodes than SMBG [6,7,9,15]. It played an especially significant role in pregnant women qualified for insulin therapy [9,19]. Chen et al. underlined CGM’s role in especially detecting nocturnal hypoglycaemia in patients requiring pharmacological treatment [9]. 

There was only one study, by Zhang et al., that calculated a significantly lower number of patients with hypoglycaemic events in the CGM group (overall, 3 patients with hypoglyceamic episodes (5.45%) in CGM versus 12 patients (21.82%) in SMBG group; χ^2^ = 6.253, *p* = 0.012) [13]. Yu at el analysed hypoglycaemia further and showed significant a difference in the duration of time spent in hypoglycaemia, with lower results in the CGM group [12]. 

### 3.2. Insulin Therapy

Five studies analysed how qualification to insulin therapy differs between the CGM and SMBG groups [6,8,18,19,20]. In three of them, it was noted that CGM is a better predictor for the initiation of antihyperglycaemic treatment [8,18,20]. Kestilä et al. found that using SMBG only leads to underestimation of the actual number of patients requiring insulin therapy [18]. In another study, it was also confirmed that CGM detects a higher number of patients who should be qualified for pharmacological treatment [8].

Two studies analysed whether CGM has an impact on insulin dosage. Paramasivan et al. conducted a randomised, controlled trial and revealed that the total insulin requirement was higher in the CGM group throughout pregnancy; however, there was no significant difference in the insulin dosage between the groups (CGM vs. control: 16.2 ± 6.4 vs. 11.8 ± 13.6 units, *p* = 0.314) [6]. An interesting outcome was demonstrated in the study by Yogev et al.; namely, the CGM group demanded 33% less long and intermediate-acting insulin, while, simultaneously, having higher (mean 20%) postprandial morning and afternoon insulin doses than the SMBG group [19]. 

### 3.3. HBA1c

HBA1c levels were analysed in six studies [5,8,10,16,18,19]. A randomized, controlled trial assessing HbA1c results in patients with GDM treated with insulin revealed significantly lower HbA1c concentration in the CGM group (CGM group: 5.2 ± 0.4% vs. SMBG group: 5.6 ± 0.6%, *p* < 0.006) [6]. Furthermore, in the CGM group, HbA1c remained unchanged, in contrast to SMBG group, in which HbA1c levels increased over the course of pregnancy. Despite the above-mentioned results in five other studies, no significant differences in HbA1c concentration between CGM and SMBG groups were observed [9,11,17,19,20].

### 3.4. Gestational Weight Gain

Gestational weight gain was analysed in three publications [14,18,20]. Two of them revealed a significantly lower increase in weight gain in the CGM group [14,20]. In addition, there was less incidence of excessive weight gain in the group using continuous glucose monitoring [14,20]. Nevertheless, the third publication, by Kestila et al., did not confirm the impact of CGM on gestational weight gain [18]. 

### 3.5. Neonatal Outcomes

Seven studies compared neonatal outcomes, and the results are not conclusive [6,11,12,13,17,18,20]. In the study by Paramasivan et al., no significant difference in newborn weight between the CGM and SMBG groups was noted (CGM: 2842.4 g ± 448.6 vs. SMBG: 2976.0 g ± 473.5; *p* = 0.311) [6]. Another two prospective studies confirmed their result [18,20]. In the study by Kestilla et al., the incidence of macrosomia was similar in both groups (*p* = 0.33) [18]. In contrast, Yu F et al. observed significantly lower neonatal weight in the CGM group (an average difference of 207 g; *p* < 0.001) and higher incidence of macrosomia or LGA in the SMBG group (*p* < 0.05) [12]. The authors also analysed other neonatal outcomes, but the results were inconclusive. There was a significantly lower incidence of neonatal hypoglycaemia and hyperbilirubinemia in the CGM group; however, NICU admission rates did not differ between the groups [12]. In four other studies, the authors showed no differences in any analysed neonatal outcomes [11,17,18,20].

## 4. Discussion

In this systematic review, we aimed to assess the efficacy of CGM on glycaemic control in GDM. Overall, the results of our review provide clear evidence for the superiority of CGM over SMBG in dysglycaemia assessment. In the majority of studies, it was shown that, in the CGM group, there was a better detection of dysglycaemia than in the SMBG group [6,7,9,12,15]. However, few studies did not confirm the statistical difference between those two methods [11,13,17]. The difference in outcomes might be the consequence of different methodologies used in the studies, including the number of patients recruited or study duration (for example, too short a period of time to reveal a statistical difference between the groups). 

An interesting outcome analysed in the review was the role of CGM in detecting nocturnal hypoglycaemia. In four studies, CGM performed better in the assessment of hypoglycaemic events [6,7,9,16]. Yu et al. revealed that CGM shortens the time spent in hypoglycaemia as compared to SMBG [12]. Moreover, the authors observed that CGM had an impact on diet control, weight monitoring and appropriate exercise. Thus, shorter time spent in hypoglycaemia was correlated with better health behaviour patterns and patient compliance. Overall, it is believed that improved nocturnal hypoglycaemia detection by CGM might have implications for better modification of GDM treatment, not only better qualification for insulin therapy, but also diet modifications [6,12]. It might play a particular role for patients requiring pharmacological treatment.

HbA1c levels, widely used as an assessment tool for patients with diabetes compliance, did not differ between the groups in almost all analysed articles [11,19,20]. Only one study noted the role of CGM in improving HbA1c levels throughout pregnancy [6]. Consequently, these outcomes might confirm that HbA1c is not the most reliable parameter used for gestational diabetes management. 

Regarding insulin therapy, almost all studies demonstrated CGM’s superiority over SMBG in predicting adequate antihyperglycaemic treatment [8,18,19,20]. Continuous glucose monitoring not only enabled better qualifications of patients for insulin therapy, but also had an impact on dose modification. CGM improved adjustments in the insulin dosage that, in consequence, could minimalize complications associated with improper treatment [20]. 

Regarding neonatal outcomes, in most of the included studies, there was no statistical difference between the CGM and SMBG groups [6,18,20]. Only one study, including over 300 patients, revealed a significantly lower incidence of LGA and lower birth weight in the CGM than in the SMBG group [12]. As a result, further studies need to be conducted to elucidate whether lack of differences in neonatal outcomes may be a consequence of methodological shortcomings. It seems likely that if continuous glucose monitoring better detects dysglycaemia and improves pharmacological treatment, it should have an impact on neonatal outcomes.

A few studies revealed the role of CGM in improving health behaviour patterns [14,20]. Zhang et al. noted its role, especially with regard to lower gestational weight gain compared to the SMBG group [14]. However, there is limited data available in other analysed studies about this maternal outcome. The possible cause of this ambiguous result might be a short period of CGM usage in the majority of the included articles (less than 7 days of measurements per patient). 

Several methodological flaws limit the internal validity of this systematic review. First, the main limitation of the analysed studies is that they included small study groups (there was only one study including >150 patients). For example, Paramasivan et al. studied the impact of CGM on maternal and neonatal outcomes with a relatively small group of patients (*n* = 25 in CGM and *n* = 25 in SMBG group) [6]. Hence, some of their results do not merge together—the study revealed the impact of continuous glucose monitoring on improving glycaemia control, but it revealed no significant differences in neonatal outcomes. Secondly, not all of the studies were randomized, controlled trials; therefore, some of the results might have been prone to recall bias. Thirdly, the periods planned for conducting the study were relatively short (median: 5 days), which may not allow the demonstration of significant differences in certain perinatal outcomes. Furthermore, there were not many studies that analysed additional maternal outcomes.

The strengths of this systematic review include study selection from five major databases and their further analysis based on clearly defined inclusion and exclusion criteria.

## 5. Conclusions

This systematic review supports the thesis that CGM is superior to SMBG in the management of dysglycaemia in GDM. Our findings suggest that CGM better detects hyper- and hypolgycaemic events and is a more appropriate predictor of qualification for insulin therapy. Therefore, these results provide justification for the idea that CGM plays an important role in glycaemia management in GDM. CGM improves the detection of fasting and postprandial hyperglycaemia. Additionally, it better assesses nocturnal hypoglycemia episodes. Improved identification of dysglycaemia allows for better patient compliance as well as decreases the rate of unnecessary interventions, including qualification for insulin therapy and further improper dose adjustment. 

On the other hand, the results for neonatal outcomes, including LGA incidence in both methods of measurement, were inconclusive. There is limited evidence that CGM improves any of the analysed neonatal outcomes. Furthermore, it will be essential to elucidate the role of CGM in changing patient health behaviour patterns. 

To conclude, more prospective studies focusing on maternal and neonatal outcomes in GDM-complicated pregnancies monitored by CGM need to be conducted to support the existing evidence and to solve the inconclusive findings.

## Figures and Tables

**Figure 1 jcm-11-02932-f001:**
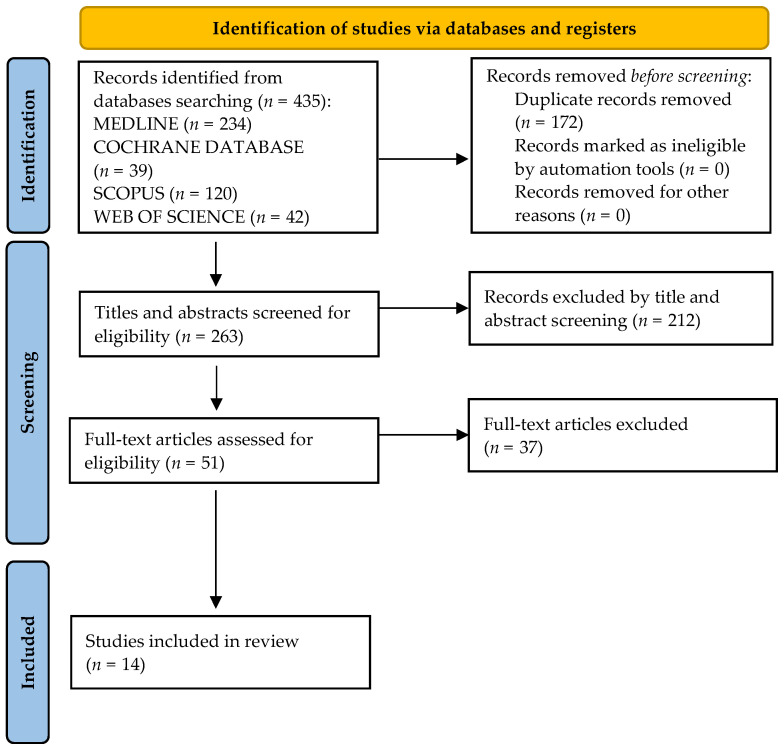
PRISMA flow diagram.

**Table 1 jcm-11-02932-t001:** Inclusion and exclusion criteria for the systematic review.

Inclusion Criteria	Exclusion Criteria
Randomized controlled trials and observational studies	Case reports, review articles, editorial comments
Human studies Studies in English	Animal studies Studies in different languages than English

**Table 2 jcm-11-02932-t002:** Characteristics of studies included in the systematic review.

Study ID	Study Design	Study Population	Type of CGM	Duration of CGM Usage	Outcome	Results
Paramasivam S et al. [6]	RCT *	57 GDM patients	iPro™ 2 Medtronic	6 days	Incidence of hypoglycaemia, insulin therapy, maternal and neonatal outcomes	Higher detection of hypoglycaemia in CGM group; no difference in other outcomes
Afandi B et al. [7]	Prospective observational study	25 GDM patients	iPro™ 2 Medtronic	5 days	Incidence of hyper- and hypoglycaemia, HbA1c level, qualification to insulin therapy	Lower incidence of hyperglycaemia and higher detection of hypoglycaemia in CGM group
Márquez-Pardo S et al. [8]	Prospective observational study	77 GDM patients	iPro™ 2 Medtronic	6 days	Incidence of hyperglycaemia, qualification to insulin therapy	Higher detection of hyperglycaemia, more qualification to insulin therapy in CGM group
Chen R et al. [9]	Prospective observational study	57 GDM patients	Medtronic MiniMed	72 h	Incidence of postprandial hyperglycaemia and nocturnal hypoglycaemia; HbA1c level	Higher detection of nocturnal hypoglycaemia and postprandial hyperglycaemia in CGM group, no difference in HbA1c level between the groups
Lane AF et al. [11]	RCT	40 GDM patients	Medtronic MiniMed/iPro™ 2 Medtronic	28 days	Incidence of hyper- and hypoglycaemia, time in range, HbA1c level, maternal and neonatal outcomes	No difference between the groups
Yu F et al. [12]	Prospective cohort study	340 GDM patients	Medtronic MiniMed	72 h a week for 5 weeks	Glycaemia control, insulin therapy, maternal and neonatal outcomes	Shorter durations of hyper- and hypoglycaemia, more patients qualified to insulin therapy in CGM group; less incidence of LGA *, neonatal hypoglycaemia and hyperbilirubinemia in CGM group
Cypryk K et al. [13]	Prospective observational study	12 GDM patients, 7 patients non-GDM	Medtronic MiniMed	72 h	Glycaemia control	No difference between the groups
Zhang X et al. [14]	RCT	110 GDM patients	ISGMS * (Abbott Diabetes Care)	14 days	Incidence of hypoglycaemia, gestational weight gain, health behaviour patterns	Lower gestational weight gain, better health behaviour patterns and lower incidence of hypoglycaemia in CGM group
Buhling KJ et al. [15]	Prospective observational study	63 GDM, 17 IGT, 24 non-GDM, 9 non-pregnant patients	Medtronic MiniMed	72 h	Glycaemia control, neonatal outcomes	Higher detection of hyperglycaemia in CGM group, no difference in other outcomes between the groups
Zaharieva D et al. [16]	Prospective Observational Study	90 GDM patients	iPRO Medtronic	7 days	Incidence of hyperglycaemia	Higher detection of hyperglycaemia in CGM group
Alfadhli E et al. [17]	RCT	130 GDM patients	Guardian^®^ RT-CGMS MiniMed	3–7 days	Fasting and postprandial glycaemia, HbA1c level, insulin therapy, maternal and neonatal outcomes	No difference between the groups
Kestila K et al. [18]	RCT	73 GDM patients	Medtronic MiniMed	Mean 47.4 h	Insulin therapy, maternal and neonatal outcomes	Higher number of patients qualified for insulin therapy in CGM group; no difference in maternal and neonatal outcomes between the groups
Yogev Y et al. [19]	Prospective observational study	6 PGDM, 2 GDM patients,	Medtronic MiniMed	72 h	Glycaemia, HbA1c level, insulin therapy, maternal and neonatal outcomes	Higher detection of nocturnal hypoglycaemia and postprandial hyperglycaemia, better modification of insulin therapy in CGM group; no difference in other outcomes between the groups
Wei Q et al. [20]	RCT	106 GDM patients	Medtronic MiniMed	48–72 h	Glycaemia, HbA1c level, insulin therapy, maternal and neonatal outcomes	Higher number of patients qualified to insulin therapy, better detection of nocturnal hypoglycaemia and postprandial hyperglycaemia, less gestational weight gain in CGM group; No difference in other outcomes between the groups

* RCT = Randomised controlled trial; LGA = large for gestational age; ISGMS: instantaneous scanning glucose monitoring system.

## Data Availability

The data that support the findings of this study are available from the corresponding author on reasonable request. The data are not publicly available due to privacy or ethical restrictions.

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
