# Peer review of "Efficacy of Continuous Glucose Monitoring on Glycaemic Control in Pregnant Women with Gestational Diabetes Mellitus—A Systematic Review"

_jcm, 2022, doi:10.3390/jcm11102932_

Round 1
Reviewer 1 Report
The manuscript entitled: “Efficacy of continuous glucose monitoring on glycaemic control 2 in pregnant women with gestational diabetes mellitus – a systematic review” by Majewska et al presents a topic of clinical interest. The methods are appropriate and the findings are interesting, well presented and they offer novelty. However, it is my belief that English can be improved.
I suggest that the manuscript is published after amendments are made.
Minor comments:
- Line 38: CGM is not a specific device, but a method of glucose monitoring. Different devices can be used. I suggest that the word “method” is used instead of “device” or that it is made clear that different devices can be used.
- Line 39: I would suggest that the word “a” is added before the word “subcutaneous”.
- In line 70, I would suggest that the word “remained” is changed to “remaining”.
- In line 169, please change “glyceamic” to “glycaemic”.
- In line 186, please correct to: “HbA1c level, widely used as an assessment tool…”.
- In line 186, please, change the phrase “diabetic patients” to “patients with diabetes”.
- In line 188, please correct to “the role of CGM”.
- In line 212, I would suggest that the sentence is corrected to “First, the main limitation of the analysed studies is that they included small study groups”.
- In line 213, please add “the” before the word “impact”.
- In line 220, I would suggest that the phrase “which may not allow to demonstrate…” is changed to “which may not allow the demonstration…”.
Author Response
The manuscript entitled: “Efficacy of continuous glucose monitoring on glycaemic control 2 in pregnant women with gestational diabetes mellitus – a systematic review” by Majewska et al presents a topic of clinical interest. The methods are appropriate and the findings are interesting, well presented and they offer novelty. However, it is my belief that English can be improved.
I suggest that the manuscript is published after amendments are made.
Author response: We appreciate the reviewer's feedback. All suggested corrections have been made.
1. Line 38: CGM is not a specific device, but a method of glucose monitoring. Different devices can be used. I suggest that the word “method” is used instead of “device” or that it is made clear that different devices can be used.
Author response: The following correction has been made – word “device” was changed to “method”.
2. Line 39: I would suggest that the word “a” is added before the word “subcutaneous”.
3. In line 70, I would suggest that the word “remained” is changed to “remaining”.
4. In line 169, please change “glyceamic” to “glycaemic”.
Author response: The proposed corrections have been made.
5. In line 186, please correct to: “HbA1c level, widely used as an assessment tool…”.
Author response: Thank you for this suggestion. The sentence has been changed to: “HbA1c level, widely used as an assessment tool for patients with diabetes compliance, did not differ between the groups in almost all analysed articles.”
6. In line 186, please, change the phrase “diabetic patients” to “patients with diabetes”.
7. In line 188, please correct to “the role of CGM”.
Author response: The proposed corrections have been made.
8. In line 212, I would suggest that the sentence is corrected to “First, the main limitation of the analysed studies is that they included small study groups”.
Author response: As suggested by the reviewer, the sentence has been changed to: “First, the main limitation of the analysed studies is that they included small study groups”.
9. In line 213, please add “the” before the word “impact”.
Author response: The proposed correction has been made.
10. In line 220, I would suggest that the phrase “which may not allow to demonstrate…” is changed to “which may not allow the demonstration…”.
Author response: Thank you for pointing this out. The sentence has been changed to: “Thirdly, the periods planned for conducting the study were relatively short (median: 5 days), which may not allow the demonstration of significant differences in certain perinatal outcomes.”
Reviewer 2 Report
The authors present a paper entitled "Efficacy of continuous glucose monitoring on glycaemic control in pregnant women with gestational diabetes mellitus – a systematic review". Despite the relevance of the topic, some concerns must be addressed.
Minor concerns:
- Please clarify the meaning of the paragraph starting at line 74.
- Do not use acronims without specifying their meaning.
Author Response
The authors present a paper entitled "Efficacy of continuous glucose monitoring on glycaemic control in pregnant women with gestational diabetes mellitus – a systematic review". Despite the relevance of the topic, some concerns must be addressed.
Author response: Thank you for all of the suggestions. We have made the following corrections to the manuscript.
1. Please clarify the meaning of the paragraph starting at line 74.
Author response: Thank you for pointing this out. We did not not perform meta-analysis due to different CGM devices used in collected studies and other methodological differences, among others duration of study (from 48 hours to 28 days) that could increase the risk of bias. Another difference that had an impact on our decision was study population, that differed among articles. Therefore, we decided to not perform meta-analysis.
Based on suggestion, the sentence has been changed to: “Due to heterogeneity in terms of device for continuous glucose monitoring, study duration and number of patients between the included articles, no meta-analysis was performed.”
2. Do not use acronims without specifying their meaning.
Author response: As suggested by the reviewer, we have specified meaning of acronims:
- PROSPERO: The International Prospective Register of Systematic Reviews (Materials and Methods, line 60)
- RCT: Randomised controlled trial (Table 2)
- LGA: Large for gestational age (Table 2)
- ISGMS: Instantaneous scanning glucose monitoring system (Table 2)